# Evolution of Barrett's esophagus through space and time at single-crypt and whole-biopsy levels

Pierre Martinez[1,2], Diego Mallo [3], Thomas G. Paulson[4], Xiaohong Li[4], Carissa A. Sanchez[4], Brian J. Reid[4,5], Trevor A. Graham [1], Mary K. Kuhner[5] & Carlo C. Maley [3,6]

The low risk of progression of Barrett's esophagus (BE) to esophageal adenocarcinoma can lead to over-diagnosis and over-treatment of BE patients. This may be addressed through a better understanding of the dynamics surrounding BE malignant progression. Although genetic diversity has been characterized as a marker of malignant development, it is still unclear how BE arises and develops. Here we uncover the evolutionary dynamics of BE at crypt and biopsy levels in eight individuals, including four patients that experienced malignant progression. We assay eight individual crypts and the remaining epithelium by SNP array for each of 6–11 biopsies over 2 time points per patient (358 samples in total). Our results indicate that most Barrett's segments are clonal, with similar number and inferred rates of alterations observed for crypts and biopsies. Divergence correlates with geographical location, being higher near the gastro-esophageal junction. Relaxed clock analyses show that genomic instability precedes and is enhanced by genome doubling. These results shed light on the clinically relevant evolutionary dynamics of BE.

---

[1] Evolution and Cancer Laboratory, Barts Cancer Institute, Queen Mary University of London, Charterhouse Square, London, EC1M 6BQ, UK. [2] Université de Lyon, Université Claude Bernard Lyon 1, INSERM 1052, CNRS 5286, Centre Léon Bérard, Cancer Research Center of Lyon, Lyon Cedex 08, 69373, France. [3] Biodesign Center for Personalized Diagnostics, Biodesign Institute, Arizona State University, Tempe, Arizona 85287, USA. [4] Divisions of Human Biology and Public Health Sciences, Fred Hutchinson Cancer Research Center, Seattle, Washington 98109-1024, USA. [5] Department of Genome Sciences, University of Washington, Seattle, Washington 98195-5065, USA. [6] School of Life Sciences, Arizona State University, Tempe, Arizona 85287, USA. Trevor A. Graham, Mary K. Kuhner and Carlo C. Maley jointly supervised this work. Correspondence and requests for materials should be addressed to C.C.M. (email: maley@asu.edu)

Barrett's esophagus (BE) is a neoplastic lesion of the eso-phagus that predisposes to esophageal adenocarcinoma (EAC)[1]. It is an ideal model for studying the dynamics of somatic evolution, because the standard of care requires long-itudinal and multi-region sampling, cataloging evolution across both space and time. Overall, the risk of progression to EAC is low: in individuals without dysplasia the annual risk is < 0.5%[2,3] and the majority of individuals with BE will never develop EAC. There is thus an acute need to avoid both over-diagnosis and over-treatment of cancer risk in non-progressors[4], and to enable earlier detection in progressors. Measuring the dynamics of progression can address these problems.

In BE, the normal squamous lining of the esophagus is replaced by columnar epithelium organized into clonally derived struc-tures resembling crypts or glands[5]. Although their architecture differs from colonic crypts, we refer to these structures as "crypts" hereafter for simplicity. The small number of stem cells present in each crypt[6,7] is thought to be rapidly homogenized by genetic drift and/or clonal selection; thus, crypts can reasonably be considered the basic units of selection in BE. Previous analyses of individual crypts have been restricted to a single time point and only a few loci per crypt[8], whereas most other studies have analyzed whole biopsies, comprising hundreds of crypts; virtually, everything we know about the evolutionary dynamics of neo-plastic progression in BE is based on studies of whole biopsies.

Genotyping of Barrett's biopsies reveals extensive somatic chromosomal aberrations (SCAs)[9–11] and point mutations[12–16]. Genome doubling (GD) and high levels of SCA were detectable in most individuals who later developed EAC 4 years before pro-gression, whereas SCA levels remained low in most non-progressors[11]. Genetic diversity (analogous to intra-tumor het-erogeneity in the context of cancerous lesions) proved to be a potent and promising marker of malignant development[17–19], yet the best strategies (in terms of both the spatial sampling and genomic analysis) to quantify diversity are unknown. Moreover, the clonal evolutionary dynamics underlying progression to cancer remain poorly characterized. Most studies have provided limited spatial resolution and it is still unclear both how BE first arises in the lower esophagus and how clonal populations develop and spread in the metaplastic tissue[15,20,21]. Spatially and geneti-cally distinct clones can all have dysplastic potential within a BE segment[13]. Clones with few alterations are still present late in progression in most cases[22], showing that genetically unstable clones do not expand to fill the entire BE segment. Furthermore, genetic diversity appears to remain stable over time, owing to a dynamic equilibrium of clones appearing and disappearing[19].

The underlying rate of SCA events in progressors and non-progressors has not been clearly determined. We previously used phylogenetic methods on whole biopsies and found a low SCA mutation rate in BE[22], consistent with a low SCA burden in those biopsies[11]. However, whole-biopsy analyses miss alterations that are confined to one or a few crypts and combine alterations present in different crypt subpopulations, which can bias the results[23]. An apparent low mutation rate in whole biopsies might be explained by a low crypt mutation rate, a low clonal expansion rate, or both. Single-crypt analyses can distinguish between these alternative hypotheses, providing evidence on the dynamics and mode of progression from BE to EAC.

In this study, we use single-nucleotide polymorphism (SNP) arrays to analyze the genomes and evolutionary relationships of multiple individual crypts and biopsies of known geographic location within the BE segments of four patients who progressed to EAC and four patients who did not progress during at least 6 years of surveillance (range: 6.1–7.6 years). We address five open questions concerning the evolutionary dynamics and neoplastic progression of BE: (1) Is the BE tissue clonal, deriving from a single altered ancestral cell? (2) Is the apparent low mutation rate at the biopsy level due to a low mutation rate or low clonal expansion rate at the crypt level? (3) Are clonal expansions common, creating a correlation between physical and genetic distances between samples? (4) Where does the BE segment originate? (5) Are there dramatic changes in the mutation (here SCA) rate during progression, leading to the evolution of mutator clones? Our findings shed new light on the evolutionary dynamics of BE and we highlight how they impact the clinical surveillance of the condition.

## Results

**Patient data sampled over time and space.** We analyzed samples from two time points for each patient, separated by a mean of 79 months (range 73–91) for non-progressors and 30 months (range 3–74) for progressors. Throughout these results, pro-gressors are indicated by -P and non-progressors by -NP appended to the patient number. For all patients, we analyzed three endoscopic biopsies at the first time point. For non-pro-gressors, an additional three biopsies were analyzed at the second time point. For progressors, eight biopsies were excised and analyzed from surgical resection specimens (see Fig. 1 for full description). The epithelium was purified by treating the biopsies with EDTA and then separating the epithelium from the stroma. This yielded a total of 48 crypts and 6 whole biopsy epitheliums (hereafter referred to as biopsies) from non-progressors, as well as 88 crypts and 11 biopsies from progressors. All samples were assayed with Illumina 2.5 M SNP arrays. The data from single crypt samples were noisier than data from whole biopsies. This limited us to reliably detecting lesions that were at least 1 Mb and, after quality control and further filtering of the data, copy number profiles were produced for a total of 358/612 samples (9–72 per patient, Supplementary Figs. 1–8). Between 75 and 174 segments were reported per patient, of size varying from 1 to 138 Mb (mean: 22 Mb; median: 14 Mb; Supplementary Fig. 9). Table 1 reports the data collection and clinical characteristics of the eight patients. The quality control and segmentation procedures are available in the Supplementary Methods.

**Analysis of breakpoints.** Using joint-segmentation and allele phasing procedures (see Supplementary Methods), we defined allele-specific breakpoints and used them as genetic markers to investigate the evolution of each BE segment. Allele phasing is the process of determining, which alleles are on the same chromo-some. In this case, in order to reconstruct the cell lineages, it is important to know whether two crypts/biopsies gained or lost the same allele, implying common ancestry, or different alleles, implying there were two independent genetic alterations. We compared genomic profiles from individual crypts to the whole epithelium from which they were isolated. There were on average 14 (range 0–65) breakpoints per crypt and $3.5 \pm 5.0$ ($23 \pm 30\%$) of those breakpoints in a crypt were not detected in the corre-sponding biopsy (range: 0–29; 0–100%; Supplementary Fig. 10). Most breakpoints found in a crypt were shared by other crypts in the same biopsy (and can be detected in whole-biopsy analysis): private breakpoints were a minority. Conversely, a crypt lacked an average of $3.9 \pm 7.4$ breakpoints ($14.2 \pm 22.0\%$) that were present in the biopsy from which it originated. This suggests that whole biopsies contain information that can still be missed by sampling multiple individual crypts and illustrates the degree of within-biopsy heterogeneity. However, across all patients neither the number of breakpoints nor the percentage of the genome altered differed significantly between biopsy and crypt samples (p = 0.4 and p = 0.9, respectively, Wilcoxon rank-sum test) (Sup-plementary Fig. 11). The number of breakpoints divergent

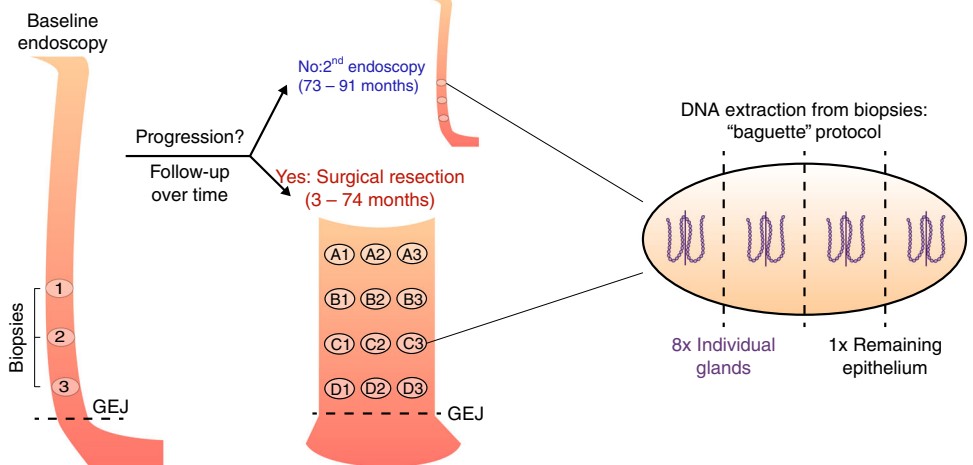

**Fig. 1** Patient samples. Progressor and non-progressor sampling over time and space. Three endoscopic biopsies were collected from all patients at the first time point. Three additional biopsies were collected from the four non-progressors during a follow-up endoscopy 73–91 months after the first time point. The distance to the GEJ was recorded for all endoscopic biopsies. Eight biopsies were collected from the surgical resection specimens of the four progressors, 3–74 months after the first time point. The location of the surgical biopsies relative to the GEJ and to each other were recorded. Each biopsy, whether endoscopic or surgical, was then split into four sections defined as baguette sections (along the long axis of the grain-of-rice-shaped biopsies), and two adjacent crypts (shown here enlarged and out of scale) were pulled from each section. The DNA of the eight individual crypts and of the remaining epithelium was extracted, yielding nine samples per biopsy

| Table 1 Patient clinical data | | | | | | | |
|---|---|---|---|---|---|---|---|
| **Patient ID** | **Sex** | **Age at 1st time point** | **Months between time points** | **Baseline diagnosis** | **Last diagnosis** | **Number of biopsies** | **Final number of samples** |
| 256-NP | M | 57 | 91.3 | Indefinite | Low grade | 6 | 13 |
| 391-P | M | 60 | 75.1 | High grade | Cancer | 11 | 62 |
| 437-NP* | M | 67 | 74.2 | High grade | High grade | 6 | 9 |
| 451-NP | M | 69 | 72.8 | High grade | Metaplasia | 6 | 21 |
| 740-P | M | 53 | 18.8 | High grade | Cancer | 11 | 72 |
| 848-P | M | 36 | 2.8 | High grade | Cancer | 11 | 63 |
| 852-P | M | 72.8 | 26.7 | High grade | Cancer | 11 | 72 |
| 911-NP | M | 73.6 | 79.3 | Low grade | Low grade | 6 | 46 |

between crypt samples and the biopsy they originated from was higher in progressors than in non-progressors (14 ± 17 vs 1 ± 4; $p < 0.001$, Wilcoxon rank-sum test; Supplementary Fig. 12). The percentage of divergent breakpoints compared to total informative breakpoints was also higher in progressors (38 ± 31% vs 12 ± 24%; $p < 0.001$, Wilcoxon rank-sum test), suggesting significantly higher heterogeneity of copy number alterations in progressors.

**Barrett's segment frequently appears clonal**. There was evidence that the BE segment was clonally derived in six out of eight patients. In four out of eight patients, one or more large genetic alterations were common to all samples (Supplementary Figs 1–8): 9p loss in patient 437-NP; 9q loss in patient 451-NP; copy-neutral LOH (cnLOH) on chromosomes 4 and 12 in patient 740-P; and 9p cnLOH in patient 911-NP (Fig. 2a). Patient 256-NP had 9p cnLOH in all samples, except for two in which no alterations were reported (Supplementary Fig. 1). Our segmentation algorithm was tuned to emphasize long segments, as short segments may be unreliable when input DNA is low (see Supplementary Methods). To increase our ability to detect shared alterations, we separately segmented and called the FHIT and WWOX loci in all eight patients (Supplementary Figs 13–28) using a segmentation procedure more sensitive to short alterations, which are expected to be frequent at these fragile-site loci[24,25]. In patient 391-P, although no obvious large clonal alteration was present in the

whole genome profiles (Fig. 2b), a ubiquitous double deletion was observed in the detailed analysis of FHIT (Fig. 2c). This implies that the BE segments of five and possibly six out of eight patients likely originated from a single cell that had acquired somatic alterations before acquiring further alterations over time. It is possible that the remaining segments also had a single-cell origin but that the originating cell did not contain any detectable SCA events.

**Maximum parsimony phylogenetic analysis**. Within-patient phylogenetic trees were computed using parsimony based on the presence of allele-specific gains or losses at breakpoint locations shared across samples (Figs 3 and 4). For each patient we constructed a geographic map of clonal development (Figs 3b and 4b, Supplementary Figs 29–34) using topographic information from endoscopic and surgical biopsy locations and color-coded phylogenetic relationships (see Supplementary Methods). Such a representation highlights genetic similarity between biopsies (Fig. 3b biopsies SS6 and SS4), genetic divergence between biopsies (Fig. 3b biopsies B2 and A3; Fig. 4b biopsies A2 and A3) and also the heterogeneity of crypt profiles within a single biopsy (Fig. 3b biopsy C3; Fig. 4b biopsy B2).

Diverse evolutionary patterns are seen in these maps. In patient 848-P (Supplementary Fig. 32), most of the heterogeneity appears to arise from a single biopsy containing eight markedly dissimilar

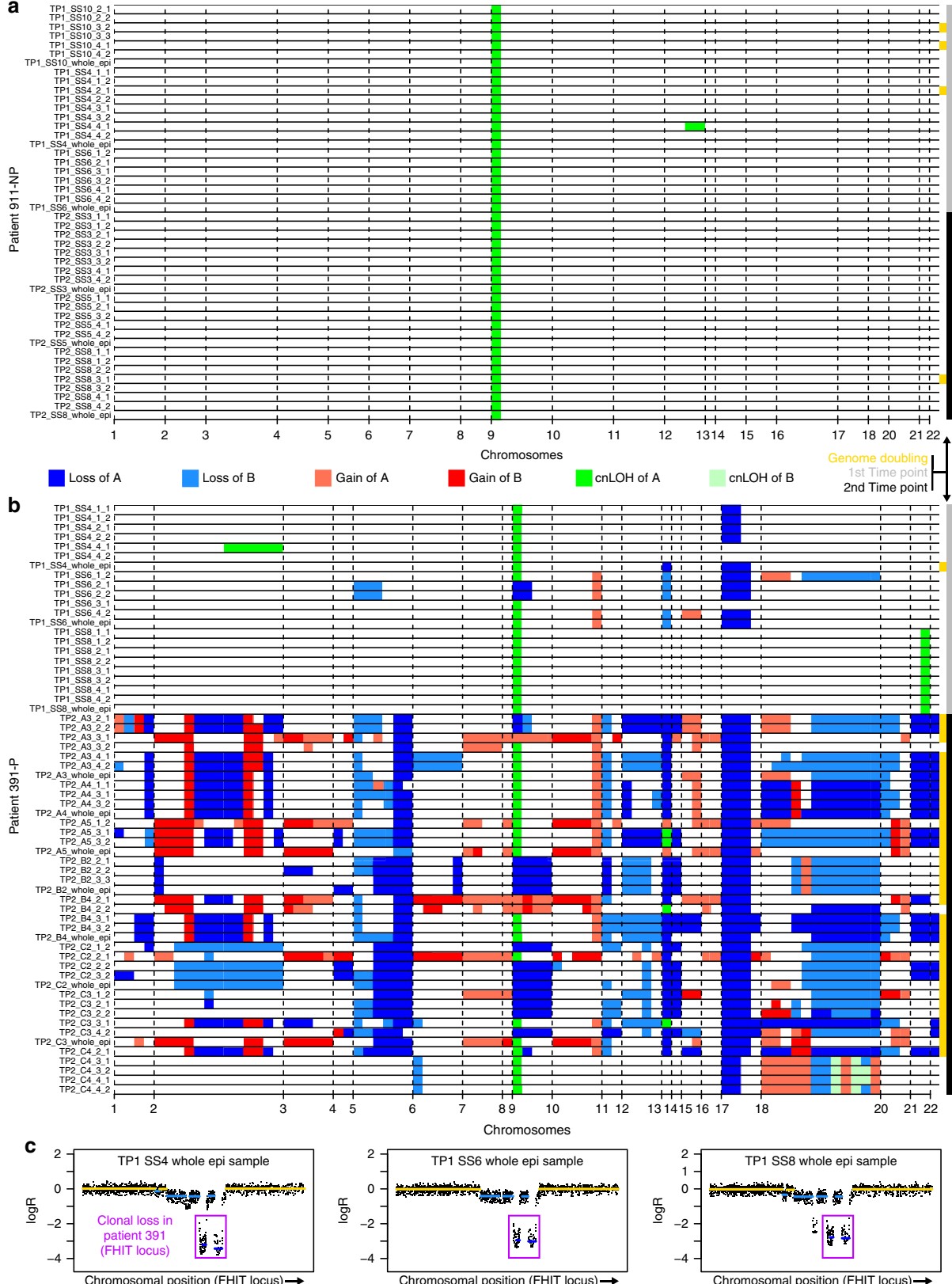

**Fig. 2** Copy number profiles. **a**, **b** Scale-free segmented maps. The copy number state of each segment is represented by equal sized bars in each sample. Yellow bars on the right side indicate whether a segment underwent genome doubling. Grey bars on the right indicate samples from the first time point, black bars those from the second time point. **a** Non-progressor patient 911-NP with 79.3 months between time points. **b** Progressor patient 391-P with 75.1 months between time points. **c** A double deletion pattern was observed at the FHIT locus across all samples from patient 391-P. Each dot in a plot is the logR value for a probe on the array and is located at the related chromosomal position on the *X* axis. Yellow segments indicate a normal copy number, light blue ones a single copy loss and dark blue ones a double loss. Only three samples are shown for illustration

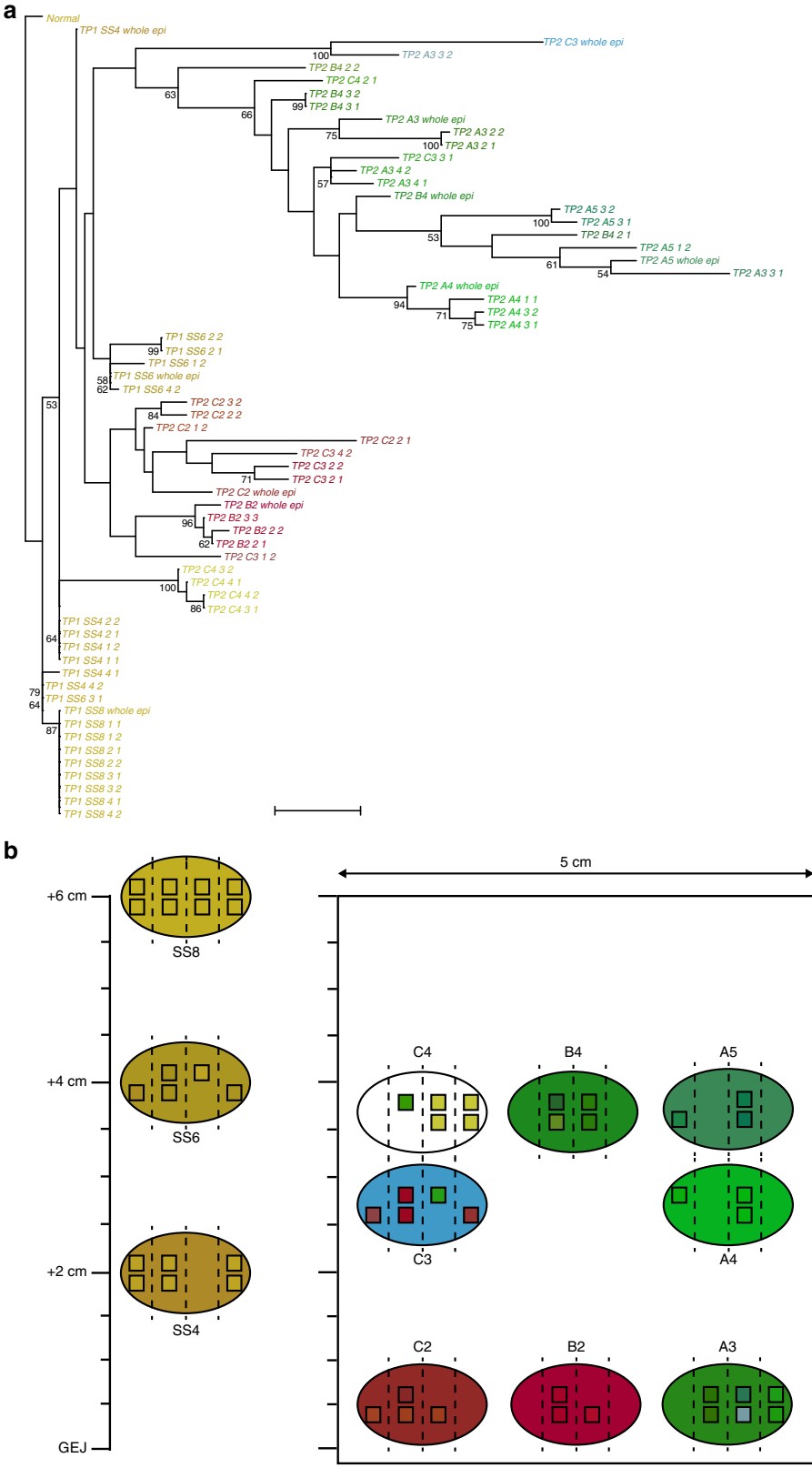

**Fig. 3** Phylogenetic tree and clonal map of patient 391-P. **a** Phylogenetic tree. Presence or absence of breakpoints are the basic genetic events used to reconstruct the tree using a parsimony algorithm. Scale bar: 10 evolutionary steps. **b** Baseline endoscopy and surgical specimen maps. Biopsies are indicated at sampling location by ovals, split into four baguette section labeled 1 to 4 from left to right. Individual crypts for which we could produce copy number profiles are indicated by squares in the relevant baguette section of the biopsy from which they originate. Colors were assigned via principal component analysis based on the presence/absence matrix of breakpoints per sample, mapping the first three components to red, green, and blue, respectively. Patient 391-P is a progressor

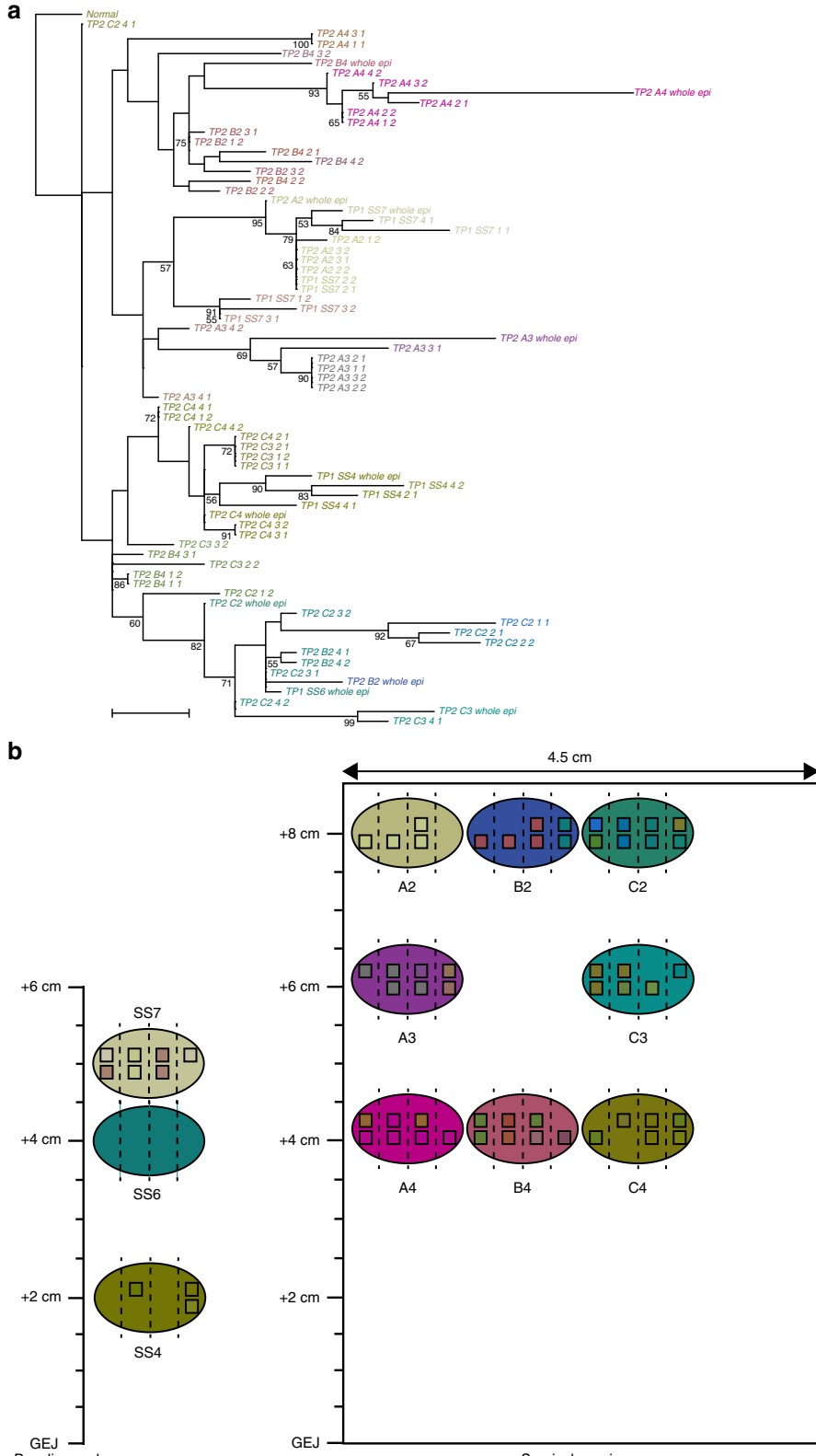

**Fig. 4** Phylogenetic tree and clonal map of patient 740-P. **a** Phylogenetic tree. Scale bar: 5 evolutionary steps. **b** Baseline endoscopy and surgical resection maps. Patient 740-P is a progressor

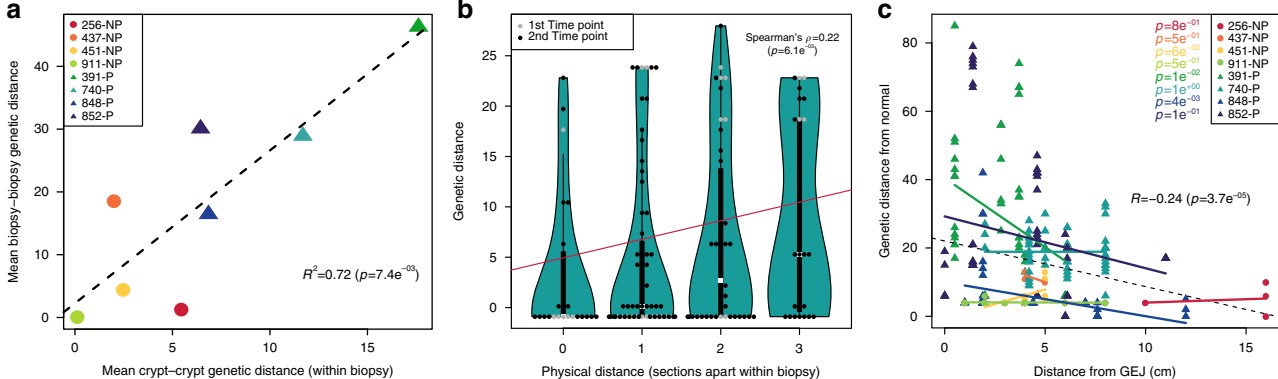

**Fig. 5** Evolutionary dynamics. **a** Correlation of evolutionary distances between different biopsies and different crypts from the same biopsy. All evolutionary distances per patient are represented by their mean. Progressors are displayed with triangles, non-progressors with circles, and each patient is indicated by a different color. $R^2$ indicates the squared Pearson's correlation coefficient used to test for significance. **b** Relation between physical and evolutionary distance in patient 852-P. Dots represent each crypt–crypt distance from the same biopsy. Grey dots are from the first time point, black ones from the second time point. Thick black bars delimit medium quartiles, thin black lines delimit 95% confidence intervals. White squares indicate the median. Correlation was tested using Spearman's non-parametric $\rho$-correlation coefficient. **c** Genetic distance between each crypt and the normal ancestral state as a function of the distance from the gastro-esophageal Junction (GEJ). Each crypt and its distance from normal is represented by a point, the color of which indicates the patient it originates from (progressors as triangles, non-progressors as circles). Colored lines indicate the linear fit for each patient, whereas the dotted black line indicates the overall fit. Individual $p$-values for Pearson's correlation coefficients per patient are indicated in the top right corner; bold font highlights statistical significance. The overall Pearson's correlation coefficient R is indicated below the plot along with its $p$-value

crypts. In contrast, in patient 852-P (Supplementary Fig. 33), the biopsies were divergent (A4, B6, and B8) but within each biopsy the sampled crypts were relatively homogeneous. These results were discordant with a second set of phylogenetic trees inferred using only fragile site data, likely reflecting the worse signal-to-noise ratio of fragile site SCAs (Supplementary Figs 35–43).

**Evolutionary distances**. We defined the evolutionary distance between two samples (crypts or biopsies) from a patient as the sum of the branch lengths connecting the samples measured on the parsimony tree. The mean evolutionary distance among a group of samples (e.g., all crypts from a biopsy) is their evolutionary diversity.

We compared evolutionary distances on the micro level (crypts within a biopsy) and the macro level (separate biopsies). In our eight patients, evolutionary diversity among crypts within a biopsy was positively correlated with diversity among biopsies (Fig. 5a: $R^2 = 0.72$, $p = 0.0074$) and with diversity among crypts from different biopsies (Supplementary Fig. 44: $R^2 = 0.71$, $p = 0.0098$). Therefore, a patient with high diversity among crypts within a biopsy will also tend to have high diversity between different biopsies.

In two patients the data were inadequate to assess this: patient 437-NP had only three usable crypt samples (two from the same biopsy) and patient 911-NP had only two lesions (one found in all samples and one in a single crypt). In five out of the six remaining patients, genetic distances between crypts from different biopsies were significantly higher than those between crypts from the same biopsy. This included all four progressors and patient 451-NP ($p < 0.05$, Wilcoxon rank-sum test, Supplementary Fig. 45). Evolutionary distances between biopsies in patient 437-NP were higher than in the other non-progressors ($p = 0.001$, Wilcoxon rank-sum test) and, after further investigation of the clinical database, it was found that this patient had undergone esophagectomy with a diagnosis of high-grade dysplasia at a different hospital, suggesting subsequent progression.

We also looked for a correlation between the physical distance among crypts within a biopsy and their evolutionary distance. In five of the six informative patients, no significant relationship was found (Supplementary Fig. 46). Physical and evolutionary

distances were positively correlated in patient 852-P, and this was still significant after correcting for the multiple patients (Spearman's $\rho = 0.22$; $p = 0.006$; corrected $p = 0.043$, Fig. 5b).

**Higher divergence near the gastro-esophageal junction**. We found that crypts nearer the gastro-esophageal junction (GEJ) had more copy number alterations ($R = -0.24$, $p < 0.001$, Fig. 5c) and displayed a higher percentage of the genome being altered ($R = -0.18$, $p = 0.002$, Supplementary Fig. 47). Our data cannot determine whether this is due to increased crypt turnover, higher mutational rate per division, or both. This finding implies that biopsy location relative to the GEJ could impact measurement of genetic diversity and mutation burden.

We used linear modeling to investigate correlations between the evolutionary distance between a pair of biopsies with progressor status, the time point at which the biopsies were taken, the physical distance between them, and the distance of the furthest biopsy from the GEJ (Table 2a). Progressor status was the most significant factor ($p = 0.002$, generalized linear model), but increased physical distance from the GEJ was also significantly associated with decreased evolutionary distance between biopsies; that is, the further a pair of biopsies were from the GEJ, the more similar they were to each other. Time point and physical distance between biopsies were not significantly correlated with genetic distance. We validated those findings by analyzing an independent cohort of 1,439 biopsies from 197 patients, in which genetic distance had been previously calculated as the percentage of 1 Mb-long genomic fragments showing a different copy number state between two biopsies (Table 2b). The results from this larger cohort confirmed the relationship between distance from the GEJ and evolutionary distance, this time with physical distance between biopsies also correlating with genetic diversity.

**SCA rate is low and similar at crypt and biopsy levels**. We used a novel Bayesian phylogenetic analysis to detect mutation rate changes during lesion evolution (Supplementary Methods). Estimated SCA mutation rates ranged from 0.005 to 0.025 events per allele copy, per locus, per year, at the crypt level and 0.003 to 0.024 at the biopsy level (Fig. 6a). Differences between estimates at the crypt and biopsy levels were small (crypt rate from 0.401 to

**Table 2 Multivariate linear models explaining the evolutionary distance between samples**

| Predictor of evolutionary distance | Estimate | SE | t-Value | p |
|---|---|---|---|---|
| *a) Single-crypt study (8 patients)* | | | | |
| (Intercept) | 14.06 | 12.27 | 1.15 | 0.5691 |
| **Progressor status (yes)** | 18.91 | 5.49 | 3.44 | **0.0017** |
| Time point (second) | 6.46 | 6.123 | 1.06 | 0.0537 |
| **Maximum distance from GEJ (cm)** | −1.48 | 0.52 | −2.84 | **0.0088** |
| Distance between biopsies (cm) | −1.03 | 0.93 | −1.11 | 0.2545 |
| *b) Large cohort (197 patients)* | | | | |
| **(Intercept)** | 4.54 | 1.50 | 3.03 | **0.0025** |
| **Progressor (yes)** | 12.75 | 0.78 | 16.30 | **$<2e^{-16}$** |
| Time point (second) | −0.25 | 0.77 | −0.33 | 0.7430 |
| **Maximum distance from GEJ (cm)** | −0.50 | 0.15 | −3.21 | **0.0014** |
| **Distance between biopsies (cm)** | 0.57 | 0.20 | 2.81 | **0.0051** |

**a**) For the eight patients in our study, genetic distances between crypts are based on phylogenetic estimates of the number of genetic events that separate the samples. **b**) For the larger cohort of Barrett's patients, the genetic distance between biopsies was calculated based on divergent copy number calls across 1 Mb windows as described in[11]. Values highlighted in bold pass the 0.05 threshold for statistical significance.

2.03 times the biopsy rate) and never statistically significant (posterior probability overlap), with posterior probability distribution overlap ranging from 0.408 to 0.789. A comparison between point estimates at the crypt level showed that progressors evolved twice as fast as non-progressors, although the difference was not significant, probably due to the small number of samples in non-progressors (mean rates 0.013 and 0.005, respectively, $p = 0.13$, Wilcoxon rank-sum test). The estimated age of the last common ancestor with an unaltered genome is an approximate estimate of the age of the Barrett's segment. These estimates varied substantially from patient to patient, and for any given patient there are wide confidence intervals on the estimated age of the segment (Supplementary Fig. 48). Despite the high degree of uncertainty of our estimates, they agree with previous results, suggesting that there is a considerable variation in BE onset times[26]. We did not find significant differences between crypt and biopsy data with respect to the estimated ages of the BE segments (posterior probability overlap). In addition, there was only weak statistical support and small differences in our estimates of effective population sizes of the evolving Barrett's cells between crypt and biopsy levels and between progressors and non-progressors (Supplementary Figs 49 and 50).

**Genome doubling**. We found that the predicted ploidies of most samples clustered around either 2 or 4, with 96% of samples having a ploidy either between 1.5 and 2.5 or above 3.5 (range per patient: 87–100%, Fig. 6b). We therefore defined samples with a predicted ploidy greater than 3 as having undergone GD. GD was detected in seven of eight patients (range: 0–54% of samples with GD per patient, median: 17%). Separate biopsies near those taken for single crypt analysis had been previously analyzed by flow cytometry for increased 4N fractions and aneuploidy[27]. The spatial distribution of the samples having flow abnormalities was similar to those determined to have undergone a GD event (Supplementary Figs. 51–58). GD occurred in both progressors and non-progressors, and was not detected significantly more often in crypts or in whole biopsy samples (all corrected $p > 0.05$, Fisher's exact test). In two patients we saw suggestive evidence of clonal expansion of GD clones. In patient 391-P, no sample from the first time point displayed GD, whereas 85% of samples from the second time point did (Fig. 7a). In patient 852-P, data were consistent with GD having occurred once and clonally expanded (Fig. 6b), whereas patient 740-P indicated multiple independent GD events throughout the BE segment. This suggests that GD can occur independently multiple times within the same BE segment and does not necessarily lead to clonal expansion.

**The evolution of SCA mutator clones**. Mutation rates evolve during neoplastic progression. To measure these changes, we carried out a random local clock analysis, which allows for changes in the mutation rate along the tree[28]. We analyzed the two patients in which we observed a clone with a doubled genome that had expanded locally (391-P and 852-P). Using crypt level data only, we estimated four SCA mutation rate changes in patient 391-P and six in patient 852-P, spanning over four orders of magnitude (Fig. 7). Both patients showed a series of increases in genomic instability (i.e., mutation rate), which preceded and then were further enhanced by the occurrence of GD. We observed a similar pattern in the analysis of the biopsy data (Supplementary Fig. 59).

## Discussion

This is the first genome-wide phylogenetic analysis of the evolutionary dynamics in BE at the level of individual crypts. The availability of two time points and geographical locations of biopsies allowed us to investigate BE development over both time and space. Copy number alteration (SCA) profiling is less precise than whole-genome sequencing, particularly to define alteration boundaries and assess the fraction of cells they affect. Although somatic mutations are less likely to be reverted to the original allele by a second mutation, SCAs such as gains are reversible and can present difficulties when inferring phylogenies. Loss-of-heterozygosity alterations are however irreversible. In addition, large SCAs are more appropriate markers for our study, as they have a key role in cancer development[29], predict progression to EA[18,19], and appear to have better potential for diversity-based prognostication than point mutations[30]. SNP arrays are a cost-effective tool to clinically investigate SCAs, as whole-genome sequencing only improves small-scale precision but greatly increases financial cost. However, SNP arrays do not reveal translocations, which prevented us from studying the role of wide-scale genome rearrangements.

In six of the eight patients, there was evidence that the BE segment derives from a single ancestral somatic cell. It is possible that the remaining BE segments may also have clonal genetic or epigenetic mutations that were missed by our SNP array approach, given that thousands of point mutations are generally present per BE genome[14]. Our extensive multi-region data are consistent with the notion that Barrett's forms from the clonal expansion of a single founder, rather than from polyclonal (trans) differentiation of multiple lineages. This is in further agreement with the contribution of CN alterations, rather than mutations, to punctuated cancer evolution[29]. However, we cannot rule out the

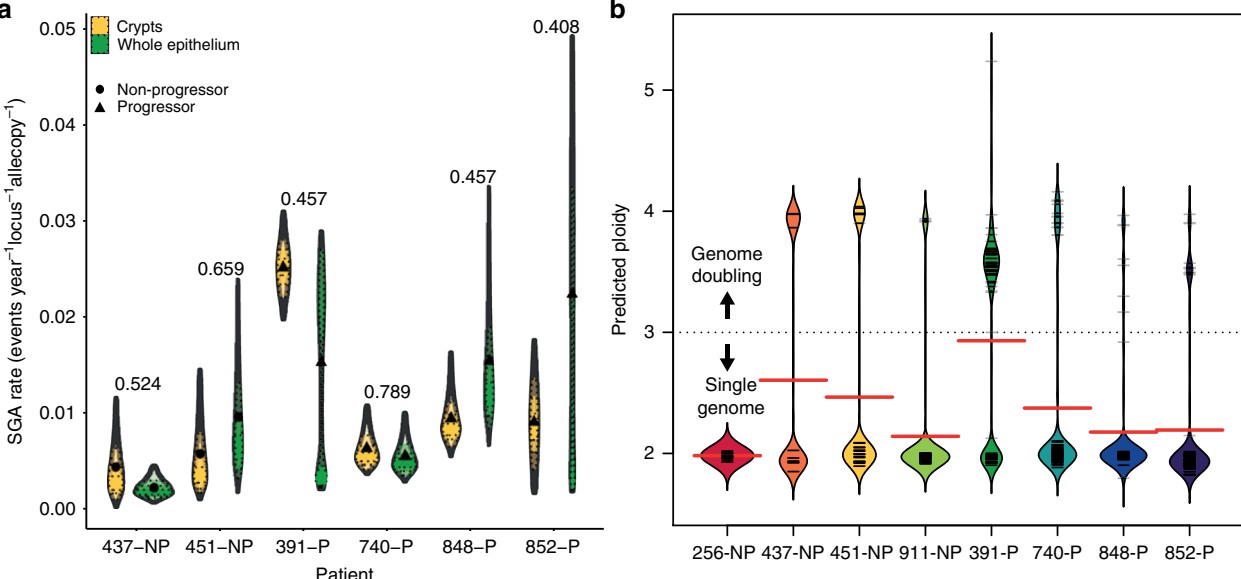

**Fig. 6** Genome instability. **a** SCA mutation rate estimation. Each violin plot corresponds to the posterior sample of the SCA mutation rate estimation for a given patient (*x* axis) and dataset (color, yellow = crypt, green = whole epithelium) included in the 95% highest posterior density (HPD) interval. The mean rate is indicated with a dot for non progressors and a triangle for progressors. Inscribed posterior samples with decreasing transparency correspond to the 75%, 50%, and 25% of the HPD interval. The probability of the two posterior distributions being the same is indicated per patient. **b** Genome Doubling. Sample ploidy as estimated by ASCAT per patient. Each horizontal black bar is an individual sample. The densities are defined and colored on a patient basis. Samples with an inferred ploidy ≥ 3 were considered to have undergone genome doubling. Horizontal red bars indicate the median of the ploidy distribution in each patient

possibility that the Barrett's segments were originally polyclonal but, before we assayed them, one clone replaced all the epithelium via early drift or selection.

Although assays of individual stem cells would provide a higher precision than crypts, the absence of bona fide BE-specific stem cell markers prevents their targeting and analysis via single-cell techniques at present. Crypts are clonally derived from distinct pools, each comprising a small number of stem cells and therefore may reasonably be considered the evolutionary units in BE. Surprisingly, we found that crypt samples had about the same number of genomic alterations as whole-biopsy samples, suggesting that biopsies provide an adequate level at which to measure evolutionary dynamics. This suggests that the stability observed in many BE segments[22] is probably due to the absence of strong selection rather than the absence of novel alterations at the crypt level. However, there were discrepancies between the crypt profile and the profile of the biopsy it originated from. This suggests that even in well-sampled regions of the esophagus some genomic alterations will be missed, which is problematic for detecting genetic modifications of malignant potential that might be present in only a fraction of the entire lesion. Reassuringly however, we find that genetic diversity at the crypt level is well reflected at the biopsy level, implying that the multiple biopsy approach efficiently measures genetic diversity. Our data thus indicate that prognostication efforts based on genetic diversity, rather than the presence of a particular genetic change, are likely to be more robust to confounding introduced by incomplete spatial sampling. The eight patients assayed in this study are from a tertiary referral cohort and presented more advanced lesions, with low- or high-grade dysplasia at baseline, than the general BE population. Most patients with BE will never develop even low-grade dysplasia. However, a recent study of a large cohort of Barrett's patients without dysplasia found that diversity measures at baseline predicted progression[19].

GD has been shown to facilitate genome instability and tumor evolution[31] and to occur close to cancer progression in BE[11].

Here we found evidence of GD in seven out of eight patients (range of 9–54% of samples in GD-positive patients), with the only exception being a non-progressor. Overlaying GD onto phylogenetic analyses suggested that it was linked to local clonal expansion in one patient (848-P) and to a nearly global expansion in another (391-P), both of them progressors. Importantly, our phylogenies show that the rise of instability and heightened SCA rates likely occurs before GD. This suggests that GD is itself a consequence of existing genomic instability: in other words, instability begets further instability. Genome doubled clones are akin to Goldschmidt's hopeful monsters[31] that appear to punctuate an otherwise largely indolent pattern of mutation accrual[22], but with the added feature that their rate of genetic alteration is increased in GD clones. The monsters appear to become 'more monstrous' over time. The fact that the same cancer pre-neoplastic lesions may evolve at different rates over time further complicates surveillance and cancer interception, with what was believed long windows of opportunity[32] possibly being shorter than first thought. Recent evidence of rapid bursts of copy number alterations punctuating cancer evolution however supports this possibility[29,33,34]. The role had by wide-scale genome rearrangements in this process cannot be assessed with SNP-array data and constitutes a topic for future investigation.

We used the geographical information at our disposal to investigate the spatial dynamics of clonal evolution in BE. At the level of individual crypts, genetic and physical distances were rarely correlated, indicating infrequent clonal expansions. Crypts from the same biopsy tended to be more closely related than crypts from different biopsies, supporting the idea that large-scale clonal expansions spanning multiple biopsies were rare, and implying that most clonal expansions occur on a scale of millimeters, not centimeters. Together with the low measured SCA rates these data indicate that, following an initial rapid colonization of the originally squamous epithelium, the crypt population evolves rather slowly, probably in the absence of strong selection. Although we did not have information on the morphological

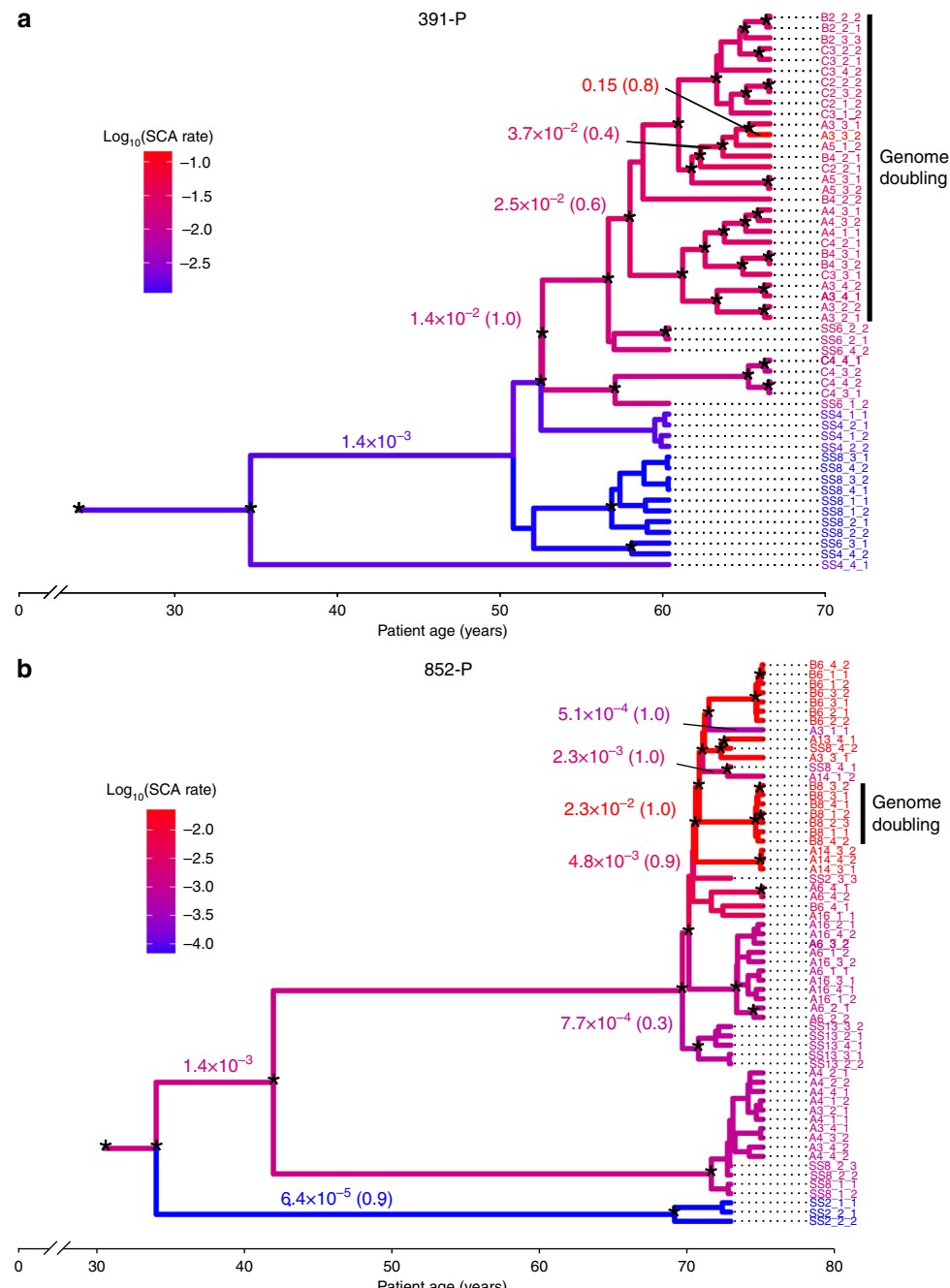

**Fig. 7** Estimated SCA mutation rate changes in patients 391-P and 852-P. Phylogenetic trees of patients 391-P **a** and 852-P **b**. Branch lengths indicate time measured in years, whereas branch colors indicate the estimated SCA rate. SCA rate changes are indicated numerically with their estimated mean rate (accompanied by their posterior probability conditional to the clade defined by their branch between parentheses). The original rate is indicated without posterior probability, as it does not constitute a rate change. Nodes with a posterior probability > 0.80 are labeled with an asterisk. The x-axis indicates years from the birth of the patient. Samples with estimated genome doubling form a monophyletic clade in both cases, indicated with a solid black line

nature of the crypts assayed in this study, it will be interesting to evaluate the association of different mucosa phenotypes with the underlying genetics of somatic evolution, in the future.

Finally, we found that the distance to the GEJ appeared to influence BE evolution, with crypts closest to the GEJ tending to show more genetic alterations and being more divergent in pairwise analyses. A possible explanation is that exposure to the components of gastric and/or bile reflux is more prominent close to the GEJ, which could increase either proliferation or DNA damage (perhaps via increased rates of epithelial wounding and repair); our data cannot distinguish between those alternative

mechanisms. This finding was validated in an independent cohort (in which genetic distance was measured differently), confirming that the location of biopsies can influence measured genetic diversity. This suggests that sampling location could bias the measurement of genetic diversity and so confound risk stratification efforts based on measures of clonal diversity[18,19]. Future work to estimate genetic diversity in BE should monitor biopsy location and either standardize the location of sampling across patients, or correct for distance from the GEJ.

Our comprehensive phylogenetic analyses of human in vivo data give new insight into the tempo and mode of somatic

evolution in BE. This broadens our knowledge of how BE develops and highlights consequences for clinical surveillance. In particular, we reveal that BE lesions likely originate from a single clone. Higher baseline instability leads to incrementally higher SCA acquisitions rates over time. This increases the probability of GD, which itself further increases SCA acquisition rates and thus the likelihood of SCA-mediated malignant progression. These data confirm the importance of assessing the evolutionary potential of BE lesions, which we show is accurately described by multiple biopsy sampling of the BE segment. However, future efforts to infer the phylogenies and clonal structure of Barrett's lesions still requires the separation of clones within biopsies[35], either through single crypt or cell analyses, or through bioinformatics deconvolution of clones[36–38]. We further highlight the important influence of spatial sampling on the measurement of evolutionary dynamics, which needs to be taken into account for evolutionary-based surveillance programs.

## Methods

**Patient cohort**. Samples were obtained from the biorepository of the Seattle Barrett's Esophagus Program (SBEP). All research participants contributing clinical data and samples for genetic analysis to this study provided written informed consent, subject to oversight by the Fred Hutchinson Cancer Research Center IRB Committee C (Reg ID 5619). Four patients who progressed to EAC during surveillance and four patients who did not progress over at least 6 years' surveillance were selected. Criteria for patient selection for progressors were availability of three biopsies from an endoscopy before detection of cancer and availability of the surgical resection specimen of the cancer and adjacent BE segment. Criteria for non-progressors were availability of three biopsies from each of two endoscopies at least 6 years apart. Progressors and non-progressors were limited to those with BE segments of 3 cm or longer and were roughly matched on segment lengths and follow-up times within the limits of available data. All samples were collected in MEM with 10% dimethyl sulfoxide, 5% fetal calf serum, 5 mmol l$^{-1}$ Hepes and frozen at − 70 °C.

For validation purposes, we also analyzed whole-biopsy data previously collected by the SBEP on an independent sample of 1,203 biopsies from 197 patients including 66 progressors and 131 non-progressors, representing a subset of the cohort described in Li et al.[10] excluding patients with inadequate sampling and patients included in the present study. These biopsies and associated blood or gastric samples had been run on Illumina 1 M OmniQuad beadchip SNP arrays for detection of SCA.

**Biopsies and single-crypt samples**. We analyzed samples from two time points for each patient. The time points were separated by a mean of 79 months (range 73–91) for non-progressors, and 25 months (range 2–75) for progressors. For all patients, we analyzed three endoscopic biopsies at the first time point. For non-progressors, an additional three endoscopic biopsies were analyzed at the second time point. For progressors, eight pseudo biopsies (called "surgical biopsies" throughout this study) were excised from surgical resection specimens consisting of the lower esophagus including the EAC tumor. In three of the four progressors, the detected EAC was microscopic and it is not known whether any of the surgical biopsies included the region of the EAC. In the fourth progressor (patient 391-P), the detected EAC was a pedunculated structure, which was not suitable for epithelial isolation and was therefore not included in the surgical biopsies.

The epithelium from each biopsy was isolated using an EDTA treatment[11]. This approach yields a specimen that is > 95% Barrett's epithelium, reducing issues caused by contamination with normal cells. Each biopsy was then divided into four "baguette" pieces (along the long axis of the grain-of-rice-shaped biopsies). Two individual crypts were isolated from each baguette. (Even in cases where the surgical biopsy may have included EAC, the well-formed crypts, which were isolated represent BE rather than EAC, as EAC tissue does not have clearly defined crypts.) The entire remaining epithelium of the biopsy was also analyzed and is referred to as the "biopsy sample" in this study. This procedure yielded a total of 48 crypts and 6 biopsies from each non-progressor, and 88 crypts and 11 biopsies from each progressor.

Genomic DNA from the epithelium of fresh frozen biopsies was isolated using PureLink Genomic DNA Mini Kit (Invitrogen/Life Technologies). Genomic DNA from individual Barrett's crypts was obtained by lysis in TE + proteinase K. 200 ng of genomic DNA was whole-genome amplified in an overnight reaction at 37 °C using multi-sample amplification master mix, and primer/neutralization mix . After incubation, the amplified DNA was fragmented with fragmentation solution, precipitated with isopropanol and precipitation mix, and resuspended in hybridization buffer (RA1). RA1 resuspended DNA was loaded onto BeadChips arrays. After overnight incubation at 48 °C, single-base extension and allele-specific staining was performed on a Teflow chamber rack system (Tecan, Maennedorf,

Switzerland). After allele-specific staining BeadChip arrays were coated with XC4/ethanol, dried for 1 h, and scanned on a iScan+ System (Illumina).

Following DNA extraction and preparation, each sample was separately analyzed on an Illumina 2.5 M OmniQuad beadchip SNP array. Gastric samples representing the normal constitutive genome were analyzed for each patient and were prepared using Puregene DNA Isolation Kits (Gentra Systems, Inc.) and quantitated with Picogreen (Quant-iT dsDNA Assay; Invitrogen)[11]. For six of eight patients, these gastric samples were obtained from the initial endoscopy. For patient 740-P the gastric sample was from the surgical resection and for patient 391-P the gastric sample was from a surveillance endoscopy taken before surgery. Table 1 reports the data collection and clinical characteristics of the eight patients.

DNA content flow cytometry ploidy data were obtained for five endoscopic biopsies and seven surgical biopsies adjacent to the biopsies used for the array assays. Flow-cytometric ploidy was assessed using previously published methods[39,40].

**Pre-processing and quality control**. Standard quality control was performed using the Illumina GenomeStudio software. Two hundred and sixteen samples did not pass the quality control (0/1 column in Supplementary Data 1) and were excluded from further analysis. logR values were corrected for GC content bias using the genomic wave correction tool of the pennCNV software suite[41].

**Bioinformatics and phylogenetics analyses**. The bioinformatics procedures and statistical tools to analyze the data are described in more detail in the Supplementary Methods (Supplementary Figs. 60–83, Supplementary Tables 1–6, and Supplementary Note 1). Briefly, as the amount of DNA per crypt sample was marginal for SNP array analysis, results were post-processed to remove areas of noisy signal shared across samples from different patients. Segments were jointly segmented based on the copynumber package[42] and the ASCAT[43] software was used for genotyping. The phangorn[44] and BEAST[45] packages were used for phylogenetic and evolutionary analyses. In order to estimate SCA mutation rates we developed a new phylogenetic method (PISCA) implemented as a BEAST 1.8 plugin (available at https://github.com/adamallo/PISCA).

**Data availability**. The original SNP-array data that supports the findings of this study are available in the NCBI GEO database (accession ID: GSE99431). All scripts used to conduct the Bayesian phylogenetic analysis are available at https://github.com/adamallo/scripts_singlecrypt for reference and reproducibility.

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

## Acknowledgements

Rumen Kostadinov shared his previous modifications to BEAST, which guided our own modifications. Joe Felsenstein and Jon Yamato assisted in developing the substitution model. This work was primarily supported by National Cancer Institute grants R01 CA140657 and P01 CA091955 (T.G.P., X.L., C.A.S., B.J.R., M.K.K., and C.C.M.). This work was also supported in part by NIH grants R01 CA149566, R01 CA170595, and R01 CA185138, as well as CDMRP Breast Cancer Research Program Award BC132057. T.A.G. was supported by Cancer Research UK (A19771). The findings, opinions, and recommendations expressed here are those of the authors and not necessarily those of the universities where the research was performed or the National Institutes of Health.

## Author contribution

P.M. designed and implemented the bioinformatics methods, performed most of the data analysis, and wrote the manuscript. D.M. and M.K.K. designed and implemented the Bayesian phylogenetic methods. D.M. performed the Bayesian analyses and wrote the manuscript sections related to those. T.G.P. performed tissue and DNA isolation and sample processing. X.L. processed the whole-biopsy SNP array data and performed its quality control. C.A.S. and B.J.R. participated in the acquisition and analysis of patient data. T.G.P., C.A.S., and B.J.R. developed and implemented the Seattle Barrett's Esophagus Project within which this study was carried out. M.K.K. and T.G.P. wrote portions of the manuscript. T.A.G., M.K.K., and C.C.M. designed the experiment, supervised the research, and edited the manuscript. All authors revised the manuscript.

## Additional information

**Competing interests:** The authors declare no competing financial interests.

