## [Peer Review File · Nature Communications]

Reviewers' comments:

Reviewer #1 (Remarks to the Author):

The manuscript describes an attempt to clarify the nature of Barrett's Esophageal development and progression. Previous studies had focused on genomic analysis at the whole bx level whereas this study included crypt level analysis. It was found, however, that the two gave similar results. The study involved tissue from 8 patients and two time points. It was well designed but, in my view failed to provide a consistent and adequately supported model for the development or progression of BE. I also don't see any important information developed that can be used for translational purposes.

Reviewer #2 (Remarks to the Author):

Martinez et al profiled the copy-number alterations in eight individuals with Barrett's esophagus, including four cancer progressors. By taking advantage of this unique sample collection system, they performed longitudinal and multi-region sampling, illustrating evolutionary dynamics across time and space (with different types of evolutionary-analysis tool). They also compared the single-crypt analyses with whole biopsies, and further showed some interesting findings: mutations were more frequently detected near the gastro-esophageal junction, genomic instability appeared to precede genome doubling, genetic diversity remains stable over time for this cancer type, and the data from single crypt samples were noisier than data from whole biopsies.

This is a very interesting report contributed by a well-respected team of Barrett's esophagus researchers. The data is solid and novel. The publication of these findings will be welcomed by the research community.

I only have a few minor questions/suggestions to further improve its presentation.

1. The advantages and disadvantages of using copy number alterations to study evolutionary dynamics need to be briefly mentioned. This might also be useful to explain the differences between the current study and recently published gene mutation profiles (Ref #13).
2. Punctuated copy number evolution was observed from aggressive subtypes of breast cancer. Can the phenotypes (slow progression and low risk of progression to EAC) be explained by the stable clonal evolution observed in the current study?
3. It should be pointed out that the current method used will not likely detect genome chaos (rapid and massive genome re-organization), one major driving force for macro-cellular evolution in many cancer types. Since cells with very high levels of polyploidy belong to a subtype of genome chaos, it would be of interest to study the issue if genome chaos can be detected from EAC (for future studies).
4. It is important to use the non-clonal event to measure genomic instability. The surprising observation that crypt samples had about the same number of genomic alterations as whole-biopsy samples could also be explained as the following: the degree of clonal aberrations are similar between the crypt and whole-biopsy; however, the non-clonal aberrations are different but less visible by the method employed. Here, single-cell analysis would be recommended for future research.
5. Please spell it out if the heterogeneity of copy number alterations is significantly different between progressors and non progressors.

Henry Heng

Reviewer #3 (Remarks to the Author):

Review of paper by Martinez and colleagues, senior author Carlo Maley.

Evolution of Barrett's oesophagus through space and time at single-crypt and whole-biopsy levels. Summary of methodology and findings: Clonal relationships in Barrett's mucosa were investigated in 8 people with Barrett's oesophagus, of whom 4 progressed to cancer, at 2 separate time points. The main conclusions (highlighted in the abstract) are:

1/ that "most" Barrett's segments were clonal in the sense that there was a set of acquired abnormalities present in every sample analysed.

2/ mutations were more frequent close to the OGJ.

3/ genomic instability appeared to precede genome doubling and clonal expansion.

4/ inter-and intra-biopsy genetic diversity were strongly correlated, and numbers of mutations and deduced mutation rates were similar in individual crypts and whole biopsies.

This study has been carried out by a consortium of investigators highly experienced in this field.

The methodology is based on the detection of copy number alterations using SNP arrays and sophisticated bioinformatics methods which I am not fully qualified to judge, although as a histopathologist with more than 20 years of the experienced in the field of Barrett's oesophagus I do hopefully have something useful to contribute.

Overall opinion: This is a comprehensive study based on a substantial body of work by able and experienced investigators, and represents a valuable contribution to our understanding of this important condition (Barrett's oesophagus).

Specific comments follow (some more important than others, not given in their order of importance).

Line 41: I think it is a mistake to crystallise the annual risk of cancer progression in Barrett's oesophagus to a single point estimate. There are just too many sources of bias and confounding (to say nothing of the impossibility of being certain that any individual is genuinely dysplasia-free). I suspect this very low figure (0.12 percent per annum) is artifactually reduced by over-ascertainment, and a more realistic figure based on meta-analysis would be about 0.5 percent per annum in people with Barrett's not known to have dysplasia (which is not the same thing as people known not to have dysplasia).

Line 46: "In Barrett's oesophagus the normal squamous lining of the oesophagus is replaced by columnar epithelium organised into clonally derived crypts". In relation to mucosal organisation the term "crypt" classically refers to the crypts of Lieberkuhn in the small intestine or the colonic crypts, both of which have a well characterised basal stem cell niche. Barrett's mucosa almost never has such a pattern of organisation, but is much more like gastric mucosa with differentiated basal glands, an intermediate region in which cell proliferation and clonal expansion characteristically are most conspicuous, and reduced proliferation with more differentiation on the mucosal surface. It is potentially misleading to apply the term "crypt" to this architecture. It is not entirely clear what the most appropriate nomenclature would be, "glands" does not seem absolutely right either because we are not necessarily talking about differentiated structures which that term might be taken to mean. Perhaps the best solution is to make a brief reference to the non-equivalents of "crypts" in Barrett's mucosa with "crypts" other contexts. This is important in the light of the statement (line 49) that "crypts can reasonably be considered the basic unit of selection and Barrett's oesophagus", which must be questionable, given that there aren't any crypts (as usually understood) in Barrett's oesophagus.

Line 58: "Genetic diversity (intra-tumour heterogeneity) proved to be a potent and promising marker

of malignant development...". Is dysplastic Barrett's mucosa a "tumour" (which this remark seems to imply)? Most people working in this field would, I think, say not.

In Barrett's oesophagus the "tumour" stage is usually adenocarcinoma, unlike the colon/rectum in which most people would accept an adenoma as a (benign) "tumour".

One could make the case that any neoplastic clonal expansion (and most people would accept the dysplastic Barrett's oesophagus does represent intraepithelial neoplasia) could be called a tumour, but the association of a tumour with a mass lesion is so strong that not to retain the distinction could be confusing.

The questions the authors wish to address are clearly stated.

They do not perhaps sufficiently emphasise that none of the 8 patients studied are representative of the general population of people with Barrett's oesophagus, most of whom do not have and will never develop even low grade dysplasia, less alone high-grade dysplasia or oesophageal adenocarcinoma. All 8 patients studied had at least low-grade dysplasia at some point, and 6 of them had baseline high-grade dysplasia. In other words, this was a cohort of people with advanced Barrett's neoplasia at the first time point of the investigation.

An interesting point is made (line 65), that "clones with few alterations are still present late in progression showing that genetically unstable clones do not expanded to fill the entire Barrett's oesophagus segment". This is concordant with morphological experience, which is reassuring.

Line 1:30: "Allele phasing techniques". A few words of explanation of precisely what this means would assist the less technical reader.

Line 148 and following: "Barrett's segments frequently appear clonal"—evidence for this in 6 of 8 patients—in the sense that "one or more large genetic alterations were common to all samples" (in 4/8 patients). Patient 256-NP was an exception (9p cnLOH not detected in 2 samples). Patient 391-P had an ubiquitous FHIT double deletion, and the authors infer a unicellular origin of the Barrett's segment in 5/6 of their patients.

But, is this not to lose sight of the fact that all of these patients have almost certainly had their Barrett's metaplasia for many years, and all have at least progressed as far as low (and in most cases high) grade dysplasia? They cannot therefore be considered representative of Barrett's oesophagus in general, nor can one conclude reliably that Barrett's oesophagus is in general monoclonal (monocellular) in origin.

Nor is this even necessarily a secure conclusion even in respect of these particular patients, because one could perfectly well envisage clonal replacement of an entire Barrett's segment long after it was initially established.

Interesting data are presented regarding phylogenetic and evolutionary distance analyses, but generalisations from these data are challenging.

The presence of greater numbers of lesions and greater diversity of lesions closer to the GOJ is an interesting and potentially important observation. This raises an issue I would like at least to see mentioned, which is the diversity of mucosal phenotype (intestinal, cardia-like, oxyntic-like) all of which can be present in Barrett's oesophagus.

There has been a widely believed (but probably incorrect) belief that only the intestinal phenotype is much associated with any elevation of cancer risk in Barrett's mucosa. There is now good evidence that non-intestinal Barrett's mucosa is also cancer-prone, and it would have been extremely interesting if the authors have been able to relate their genetic data to mucosal phenotype.

This might be especially interesting given that the authors document more numerous and more diverse genetic lesions towards the OGJ. There is reliable published evidence that cardiac-type Barrett's mucosa is more likely to be present at this location, and one wonders whether the cardiac type mucosa is particularly mutation-prone.

This is possibly of crucial importance for the epidemic of adenocarcinoma now occurring at the oesophagogastric junction even in the absence of obvious Barrett's metaplasia, and Chandrasoma's observations that cardiac type mucosa at the junction may very often itself be metaplastic (but not

usually intestinal).

Estimating the age of the Barrett's segment from mutations and possible evolution rates seems to me so dependent on a plethora of assumptions that I question its likely validity (lines 285 and following).

It was interesting to see data supporting the importance of genome doubling in the evolution of Barrett's oesophagus (lines 301 and following), and evidence for SCA mutation rate changes from which the existence of SCA mutator clones can be inferred.

Lines 343 and following: I do not think we can be sure that the Barrett segment as originally constituted, probably long before the patient came to the attention of the medical profession, was derived from a single ancestral somatic cell. Perhaps the authors do not mean to suggest this, but their remarks certainly can be so interpreted. Their data however do not exclude "polyclonal (trans)differentiation of multiple lineages" in the establishment of Barrett's metaplasia.

I do however find myself in complete agreement that "prognostication based on genetic diversity" is likely to be much more effective than prognostication based on specific genetic events, and I look forward to progress in this important but challenging area.

I also find myself entirely in agreement with the suggestion that sudden acceleration of cancer development is eminently possible and I have seen examples of this in my clinical practice over the last 20 years. Such phenomena clearly increase the challenge of cancer interception.

Line 391: "Acid mediating epithelial wounding..." It is not just acid that damages mucosa, pepsin probably plays a part and bile acids/bile salts as well.

We thank the reviewers for their helpful comments and critiques. The manuscript has been improved in the process.

Below we use the following color codes: Reviewers, Authors, "Quoted main text".

Reviewer #1 (Remarks to the Author):

The manuscript describes an attempt to clarify the nature of Barrett's Esophageous development and progression. Previous studies had focused on genomic analysis at the whole bx level whereas this study included crypt level analysis. It was found, however, that the two gave similar results. The study involved tissue from 8 patients and two time points. It was well designed but, in my view failed to provide a consistent and adequately supported model for the development or progression of BE. I also don't see any important information developed that can be used for translational purposes.

The study wasn't designed to develop a complete model of BE development or progression for translational purposes, but rather to reveal the underlying evolutionary dynamics at the crypt level. This allowed us to answer six previously-open questions:

1. Is the BE tissue clonal, deriving from a single altered ancestral cell? Yes, in most cases.
2. Is the apparent low mutation rate at the biopsy level due to a low mutation rate or low clonal expansion rate at the crypt level? We found a low mutation rate at the crypt level.
3. Are clonal expansions common, creating a correlation between physical and genetic distances between samples? No. Large clonal expansions are rare, though very small ones within a biopsy are more common.
4. Why do most esophageal cancers arise near the gastro-esophageal junction (GEJ)? We found that crypts and biopsies near the GEJ accumulated more genomic alterations than those farther away from the GEJ. This suggests that samples near the GEJ are more likely to be informative for prognosis.
5. Are there dramatic changes in the mutation (here SCA) rate during progression, leading to the evolution of mutator clones? Yes, and those mutators emerge prior to the occurrence of genome doubling.
6. Are we accurately measuring evolutionary dynamics when we only measure whole biopsies? We found similar levels of genetic diversity and mutation rates at both the crypt and the biopsy level, suggesting that assaying biopsies is adequate for characterizing somatic evolution in BE.

Reviewer #2 (Remarks to the Author):

Martinez et al profiled the copy-number alterations in eight individuals with Barrett's esophagus, including four cancer progressors. By taking advantage of this unique sample collection system, they performed longitudinal and multi-region sampling, illustrating evolutionary dynamics across time and space (with different types of evolutionary-analysis tool). They also compared the single-crypt analyses with whole biopsies, and further showed some interesting findings: mutations were more frequently detected near the gastro-esophageal junction, genomic instability appeared to precede genome doubling, genetic diversity remains stable over time for this cancer type, and the data from single crypt samples were noisier than data from whole biopsies.

This is a very interesting report contributed by a well-respected team of Barrett's esophagus researchers. The data is solid and novel. The publication of these findings will be welcomed by the research community.

We thank the reviewer for the positive comments.

I only have a few minor questions/suggestions to further improve its presentation.

1. The advantages and disadvantages of using copy number alterations to study evolutionary dynamics need to be briefly mentioned. This might also be useful to explain the differences between the current study and recently published gene mutation profiles (Ref #13).

We agree with the reviewer and have extended or discussion of the matter, including the role of copy number alterations (*changes in italics*):

"This is the first genome-wide phylogenetic analysis of the evolutionary dynamics in BE at the level of individual crypts. The availability of two time points and geographical locations of biopsies allowed us to investigate BE development over both time and space. *Copy number alteration (CNA) profiling is less precise than whole-genome sequencing, particularly to define alteration boundaries and assess the fraction of cells they affect. While somatic mutations are less likely to be reverted to the original allele by a second mutation, CNAs such as gains are reversible and can present difficulties when inferring phylogenies. Loss-of heterozygosity alterations are, however, irreversible. In addition, large CNAs are more appropriate markers for our study since they play a key role in cancer development²⁸, predict progression to EA^{17,18}, and appear to have better potential for diversity-based prognostication than point mutations²⁹. SNP-arrays are a cost-effective tool to clinically investigate CNAs since whole genome sequencing only improves small-scale precision but significantly increases financial cost. However, SNP arrays do not reveal translocations which prevented us from studying the role of wide-scale genome rearrangements.*"

2. Punctuated copy number evolution was observed from aggressive subtypes of breast cancer. Can the phenotypes (slow progression and low risk of progression to EAC) be explained by the stable clonal evolution observed in the current study?

Yes, the stable clonal evolution, due to low mutation rates and weak selection, probably explain the low risk of progression. Punctuated copy number evolution is likely to involve chromosomal catastrophes with multiple alterations co-occurring. Gao et al. indeed report on chromosomal catastrophes that

“these data are unlikely to be explained by the gradual accumulation of copy number events over time”. These catastrophes are therefore less likely to arise with the low reported CNA rates and absence of genome doubling reported in non-progressors. The surging mutational rates, appearing to precede genome doubling, will however increase the probability of catastrophic events, and therefore of potentially malignant clones arising. We address this issue in the following paragraph of our discussion (reference 29 is the breast cancer paper the reviewer mentions):

“Genome doubling (GD) has been shown to facilitate genome instability and tumor evolution³² and to occur close to cancer progression in BE¹¹. Here we found evidence of GD in 7 out of 8 patients (range of 9%-54% of samples in GD-positive patients), with the only exception being a non-progressor. Overlaying GD onto phylogenetic analyses suggested that it was linked to local clonal expansion in one patient (848-P) and to a nearly global expansion in another (391-P), both of them progressors. Importantly, our phylogenies show that the rise of instability and heightened SCA rates likely occurs prior to genome doubling. This suggests that GD is itself a consequence of existing genomic instability: in other words, instability begets further instability. Genome doubled clones are akin to Goldschmidt’s *hopeful monsters*³⁰ that appear to punctuate an otherwise largely indolent pattern of mutation accrual²², but with the added feature that their rate of genetic alteration is increased in GD clones. The monsters appear to become ‘more monstrous’ over time. The fact that the same cancer pre-neoplastic lesions may evolve at different rates over time further complicates surveillance and cancer interception, with what was believed long windows of opportunity³¹ possibly being shorter than first thought. Recent evidence of rapid bursts of copy number alterations punctuating cancer evolution however supports this possibility^{29,32,33}.”

3. It should be pointed out that the current method used will not likely detect genome chaos (rapid and massive genome re-organization), one major driving force for macro-cellular evolution in many cancer types. Since cells with very high levels of polyploidy belong to a subtype of genome chaos, it would be of interest to study the issue if genome chaos can be detected from EAC (for future studies).

We agree, the SNP arrays do not reveal translocations, but only the copy number changes that may result. In discussion of future directions we have added the following comment:

“The role played by wide-scale genome rearrangements in this process cannot be assessed with SNP-array data, and constitutes a topic for future investigation.”

4. It is important to use the non-clonal event to measure genomic instability. The surprising observation that crypt samples had about the same number of genomic alterations as whole-biopsy samples could also be explained as the following: the degree of clonal aberrations are similar between the crypt and whole-biopsy; however, the non-clonal aberrations are different but less visible by the method employed. Here, single-cell analysis would be recommended for future research.

It is true that we are still missing lesions in the very tips of the cell lineages, that are non-clonal in the crypts. This is a complicated issue, because most of the cells in a crypt are not stem cells, and so have little evolutionary potential. Unfortunately, we currently lack reliable markers of stem cells in BE, and so, single cell sequencing of stem cells is currently not feasible. The advantage of assaying single crypts is that they survey mutations in the stem cells that are amplified by proliferation in the crypt. We have discussed this in the following sentences added to the discussion:

“Although assays of individual stem cells would provide a higher precision than crypts, the absence of bona fide BE-specific stem cell markers prevents their targeting and analysis via single-cell techniques at present. Crypts are clonally derived from distinct pools, each comprising a small number of stem cells, and therefore may reasonably be considered the evolutionary units in BE. Surprisingly, we found that crypt samples had about the same number of genomic alterations as whole-biopsy samples, suggesting that biopsies provide an adequate level at which to measure evolutionary dynamics.”

5. Please spell it out if the heterogeneity of copy number alterations is significantly different between progressors and non progressors.

We updated the corresponding paragraph from the results section (*changes in italics*):

“The percentage of divergent breakpoints compared to total informative breakpoints was also higher in progressors (38% ± 31% vs 12% ± 24%; p<0.001, Wilcoxon rank sum test), suggesting significantly higher heterogeneity of copy number alterations in progressors.”

Henry Heng

Reviewer #3 (Remarks to the Author):

Review of paper by Martinez and colleagues, senior author Carlo Maley.

Evolution of Barrett's oesophagus through space and time at single-crypt and whole-biopsy levels.

Summary of methodology and findings: Clonal relationships in Barrett's mucosa were investigated in 8 people with Barrett's oesophagus, of whom 4 progressed to cancer, at 2 separate time points. The main conclusions (highlighted in the abstract) are:

1/ that "most" Barrett's segments were clonal in the sense that there was a set of acquired abnormalities present in every sample analysed.

2/ mutations were more frequent close to the OGJ.

3/ genomic instability appeared to precede genome doubling and clonal expansion.

4/ inter-and intra-biopsy genetic diversity were strongly correlated, and numbers of mutations and deduced mutation rates were similar in individual crypts and whole biopsies.

This study has been carried out by a consortium of investigators highly experienced in this field.

The methodology is based on the detection of copy number alterations using SNP arrays and sophisticated bioinformatics methods which I am not fully qualified to judge, although as a histopathologist with more than 20 years of the experienced in the field of Barrett's oesophagus I do hopefully have something useful to contribute.

Overall opinion: This is a comprehensive study based on a substantial body of work by able and experienced investigators, and represents a valuable contribution to our understanding of this important condition (Barrett's oesophagus).

We thank the reviewer for the positive comments.

Specific comments follow (some more important than others, not given in their order of importance).

Line 41: I think it is a mistake to crystallised the annual risk of cancer progression in Barrett's oesophagus to a single point estimate. There are just too many sources of bias and confounding (to say nothing of the impossibility are being certain that any individual is genuinely dysplasia-free). I suspect this very low figure (0.12 percent per annum) is artifactually reduced by over-ascertainment, and a more realistic figure based on meta-analysis would be about 0.5 percent per annum in people with Barrett's not known to have dysplasia (which is not the same thing is people known not to have dysplasia).

We have amended the sentence and added a recent meta-analysis to the following (*changes in italics*):

"Overall, the risk of progression to EAC is low: in individuals without dysplasia the annual risk is *less than 0.5%^{2,3}*, and the majority of individuals with BE will never develop EAC."

Line 46: "In Barrett's oesophagus the normal squamous lining of the oesophagus is replaced by columnar epithelium organised into clonally derived crypts". In relation to mucosal organisation the term "crypt" classically refers to the crypts of Lieberkuhn in the small intestine or the colonic crypts, both of which

have a well characterised basal stem cell niche. Barrett's mucosa almost never has such a pattern of organisation, but is much more like gastric mucosa with differentiated basal glands, an intermediate region in which cell proliferation and clonal expansion characteristically are most conspicuous, and reduced proliferation with more differentiation on the mucosal surface. It is potentially misleading to apply the term "crypt" to this architecture. It is not entirely clear what the most appropriate nomenclature would be, "glands" does not seem absolutely right either because we are not necessarily talking about differentiated structures which that term might be taken to mean.

Perhaps the best solution is to make a brief reference to the non-equivalents of "crypts" in Barrett's mucosa with "crypts" other contexts. This is important in the light of the statement (line 49) that "crypts can reasonably be considered the basic unit of selection and Barrett's oesophagus", which must be questionable, given that there aren't any crypts (as usually understood) in Barrett's oesophagus.

Although the nomenclature relative to the crypt- or gland-like architecture observed in Barrett's is important, we do not have morphological data to characterize the crypts we have analyzed. We agree that clarification of what is meant in the present article by the term crypt is required and have amended the corresponding paragraph (*changes in italics*):

"In BE the normal squamous lining of the esophagus is replaced by columnar epithelium organized into clonally derived *structures resembling crypts or glands*⁴. *Although their architecture differs from colonic crypts, we refer to these structures as "crypts" hereafter for simplicity.* The small number of stem cells present in each crypt^{5,6} are thought to be rapidly homogenized by genetic drift and/or clonal selection; thus, crypts can reasonably be considered the basic units of selection in BE."

Line 58: "Genetic diversity (intra-tumour heterogeneity) proved to be a potent and promising marker of malignant development...". Is dysplastic Barrett's mucosa a "tumour" (which this remark seems to imply)? Most people working in this field would, I think, say not.

In Barrett's oesophagus the "tumour" stage is usually adenocarcinoma, unlike the colon/rectum in which most people would accept an adenoma as a (benign) "tumour".

One could make the case that any neoplastic clonal expansion (and most people would accept the dysplastic Barrett's oesophagus does represent intraepithelial neoplasia) could be called a tumour, but the association of a tumour with a mass lesion is so strong that not to retain the distinction could be confusing.

We agree that the sentence could have introduced some confusion. We have updated it to the following:

"Genetic diversity (*analogous to intra-tumor heterogeneity in the context of cancerous lesions*)".

The questions the authors wish to address are clearly stated.

They do not perhaps sufficiently emphasise that none of the 8 patients studied are representative of the general population of people with Barrett's oesophagus, most of whom do not have and will never develop even low grade dysplasia, less alone high-grade dysplasia or oesophageal adenocarcinoma. All 8 patients studied had at least low-grade dysplasia at some point, and 6 of them had baseline high-grade dysplasia. In other words, this was a cohort of people with advanced Barrett's neoplasia at the first time point of the investigation.

This is a good point. It is reassuring that our previous study of a large cohort of Barrett's patients without any dysplasia found that (single cell FISH) diversity measures predicted progression (ref 19). However, to acknowledge the fact that the patients assayed in this study have more advanced BE neoplasia than the general BE population, we have added the following sentences to the discussion:

"The 8 patients assayed in this study are from a tertiary referral cohort and presented more advanced lesions, with low or high grade dysplasia at baseline, than the general BE population. Most patients with BE will never develop even low grade dysplasia. However, a recent study of a large cohort of Barrett's patients without dysplasia found that diversity measures at baseline predicted progression¹⁹."

An interesting point is made (line 65), that "clones with few alterations are still present late in progression showing that genetically unstable clones do not expanded to fill the entire Barrett's oesophagus segment". This is concordant with morphological experience, which is reassuring.

Line 1:30: "Allele phasing techniques". A few words of explanation of precisely what this means would assist the less technical reader.

We agree and have added these sentences:

"Allele phasing is the process of determining which alleles are on the same chromosome. In this case, in order to reconstruct the cell lineages, it is important to know if two crypts/biopsies gained or lost the same allele, implying common ancestry, or different alleles, implying there were two independent genetic alterations."

Line 148 and following: "Barrett's segments frequently appear clonal"—evidence for this in 6 of 8 patients—in the sense that "one or more large genetic alterations were common to all samples" (in 4/8 patients). Patient 256—NP was an exception (9p cnLOH not detected in 2 samples). Patient 391—P had an ubiquitous FHIT double deletion, and the authors infer a unicellular origin of the Barrett's segment in 5/6 of their patients.

But, is this not to lose sight of the fact that all of these patients have almost certainly had their Barrett's metaplasia for many years, and all have at least progressed as far as low (and in most cases high) grade dysplasia? They cannot therefore be considered representative of Barrett's oesophagus in general, nor can one conclude reliably that Barrett's oesophagus is in general monoclonal (monocellular) in origin.

Nor is this even necessarily a secure conclusion even in respect of these particular patients, because one could perfectly well envisage clonal replacement of an entire Barrett's segment long after it was initially established.

The reviewer has a fair point. The crypts in our study might not be initiated by a single clone, but by the time we assayed it, they were clonal. We have added the following sentence (*in italics*) to the corresponding paragraph:

"Our extensive multi-region data are consistent with the notion that Barrett's forms from the clonal expansion of a single founder, rather than from polyclonal (trans)differentiation of multiple lineages. This is in further agreement with the contribution of CN alterations, rather than mutations, to punctuated cancer evolution²⁹. However, we cannot rule out the possibility that the Barrett's segments were originally polyclonal but, before we assayed them, one clone replaced all the epithelium via early drift or selection."

Interesting data are presented regarding phylogenetic and evolutionary distance analyses, but generalisations from these data are challenging.

The presence of greater numbers of lesions and greater diversity of lesions closer to the GOJ is an interesting and potentially important observation. This raises an issue I would like at least to see mentioned, which is the diversity of mucosal phenotype (intestinal, cardia-like, oxyntic-like) all of which can be present in Barrett's oesophagus.

There has been a widely believed (but probably incorrect) belief that only the intestinal phenotype is much associated with any elevation of cancer risk in Barrett's mucosa. There is now good evidence that non-intestinal Barrett's mucosa is also cancer-prone, and it would have been extremely interesting if the authors have been able to relate their genetic data to mucosal phenotype.

This might be especially interesting given that the authors document more numerous and more diverse genetic lesions towards the OGJ. There is reliable published evidence that cardiac-type Barrett's mucosa is more likely to be present at this location, and one wonders whether the cardiac type mucosa is particularly mutation-prone.

This is possibly of crucial importance for the epidemic of adenocarcinoma now occurring at the oesophagogastric junction even in the absence of obvious Barrett's metaplasia, and Chandrasoma's observations that cardiac type mucosa at the junction may very often itself be metaplastic (but not usually intestinal).

We agree with the reviewer that different type of mucosa can imply different progression risks. Unfortunately, we do not have morphological and pathological information on the crypts that were assayed. This would be an interesting topic of future investigation as we now indicate in the discussion:

"Together with the low measured SCA rates these data indicate that, following an initial rapid colonization of the originally squamous epithelium, the crypt population evolves rather slowly, probably in the absence of strong selection. Although we did not have information on the morphological nature of the crypts assayed in this study, it will be interesting to evaluate the association of different mucosa phenotypes with the underlying genetics of somatic evolution, in the future."

Estimating the age of the Barrett's segment from mutations and possible evolution rates seems to me so dependent on a plethora of assumptions that I question its likely validity (lines 285 and following).

We agree with this observation and acknowledged the issue by updating the corresponding results section to:

"The estimated age of the last common ancestor with an unaltered genome is an approximate estimate of the age of the Barrett's segment. These estimates varied substantially from patient to patient, and for any given patient there are wide confidence intervals on the estimated age of the segment (Supplementary Fig. 49). Despite the high degree of uncertainty of our estimates, they agree with previous results suggesting that there is a considerable variation in BE onset times²⁵."

It was interesting to see data supporting the importance of genome doubling in the evolution of Barrett's oesophagus (lines 301 and following), and evidence for SCA mutation rate changes from which the existence of SCA mutator clones can be inferred.

Lines 343 and following: I do not think we can be sure that the Barrett segment as originally constituted, probably long before the patient came to the attention of the medical profession, was derived from a single ancestral somatic cell. Perhaps the authors do not mean to suggest this, but their remarks certainly can be so interpreted. Their data however do not exclude "polyclonal (trans)differentiation of multiple lineages' in the establishment of Barrett's metaplasia.

As for the earlier comment on the matter, we have added the following sentence to the corresponding paragraph:

"However, we cannot rule out the possibility that the Barrett's segments were originally polyclonal but, before we assayed them, one clone replaced all the epithelium via early drift or selection."

I do however find myself in complete agreement that "prognostication based on genetic diversity" is likely to be much more effective than prognostication based on specific genetic events, and I look forward to progress in this important but challenging area.

Thank you for the encouragement.

I also find myself entirely in agreement with the suggestion that sudden acceleration of cancer development is eminently possible and I have seen examples of this in my clinical practice over the last 20 years. Such phenomena clearly increase the challenge of cancer interception.

Line 391: "Acid mediating epithelial wounding..." It is not just acid that damages mucosa, pepsin probably plays a part and bile acids/bile salts as well.

We have updated the corresponding section to a less acid-specific sentence:

"A possible explanation is that exposure to the components of gastric and/or bile reflux is more prominent close to the GEJ, which could increase either proliferation or DNA damage (perhaps via increased rates of epithelial wounding and repair); our data cannot distinguish between those alternative mechanisms."

REVIEWERS' COMMENTS:

Reviewer #1 (Remarks to the Author):

Placing these experimental findings in a broader context of how they contribute to understanding the observed prognosis and pathogenesis of BE, may help the reader appreciate the significance of the findings.

Reviewer #2 (Remarks to the Author):

With all these modifications, I believe that this MS is now ready to be accepted.

Reviewer #3 (Remarks to the Author):

I consider that the authors rebuttal / response to comments is balanced and appropriate and I would personally now recommend publication.

Answer to reviewers

We use the following color code: **Reviewer comments in red**, our response in black.

Reviewer #1 (Remarks to the Author):

Placing these experimental findings in a broader context of how they contribute to understanding the observed prognosis and pathogenesis of BE, may help the reader appreciate the significance of the findings.

We have updated the last paragraph of the discussion to highlight how our study contributes to BE prognosis and pathogenesis. We replaced the former paragraph

“Our comprehensive phylogenetic analyses of human in vivo data give new insight into the tempo and mode of somatic evolution in Barrett's Esophagus. In particular, we reveal the pattern of mutation rate evolution during neoplastic progression and the important influence of spatial sampling on the measurement of evolutionary dynamics.”

With the more explicit one below:

“Our comprehensive phylogenetic analyses of human in vivo data give new insight into the tempo and mode of somatic evolution in Barrett's Esophagus. This broadens our knowledge of how BE develops and highlights consequences for clinical surveillance. In particular, we reveal that BE lesions likely originate from a single clone. Higher baseline instability leads to incrementally higher CNA acquisitions rates over time. This increases the probability of genome doubling, which itself further increases CNA acquisition rates and thus the likelihood of CNA-mediated malignant progression. These data confirm the importance of assessing the evolutionary potential of BE lesions, which we show is accurately described by multiple biopsy sampling of the BE segment. However, future efforts to infer the phylogenies and clonal structure of Barrett's lesions still requires the separation of clones within biopsies³⁵, either through single crypt or cell analyses, or through bioinformatics deconvolution of clones³⁶⁻³⁸. We further highlight the important influence of spatial sampling on the measurement of evolutionary dynamics, which needs to be taken into account for evolutionary-based surveillance programmes.”

Reviewer #2 (Remarks to the Author):

With all these modifications, I believe that this MS is now ready to be accepted.
Nothing to amend.

Reviewer #3 (Remarks to the Author):

I consider that the authors rebuttal / response to comments is balanced and appropriate and I would personally now recommend publication.

Nothing to amend.